

# Computer modelling reveals new conformers of the ATP binding loop of Na+/K+-ATPase involved in the transphosphorylation process of the sodium pump

Gracian Tejral[1,2], Bruno Sopko[3], Alois Necas[4], Wilhelm Schoner[5] and Evzen Amler[1,2]

[1] Department of Biophysics, 2nd Faculty of Medicine, Charles University Prague, Prague, Czech Republic
[2] Laboratory of Tissue Engineering, Institute of Experimental Medicine, Academy of Sciences of the Czech Republic, Prague, Czech Republic
[3] Department of Medical Chemistry and Clinical Biochemistry, 2nd Faculty of Medicine, Charles University Prague, Prague, Czech Republic
[4] Small Animal Clinic, Faculty of Veterinary Medicine, University of Veterinary and Pharmaceutical Science, Brno, Czech Republic
[5] Institute of Biochemistry and Endocrinology, University of Giessen, Giessen, Germany

Corresponding author
Evzen Amler,
evzen.amler@lfmotol.cuni.cz

## ABSTRACT

Hydrolysis of ATP by Na+/K+-ATPase, a P-Type ATPase, catalyzing active Na+ and K+ transport through cellular membranes leads transiently to a phosphorylation of its catalytical $\alpha$-subunit. Surprisingly, three-dimensional molecular structure analysis of P-type ATPases reveals that binding of ATP to the N-domain connected by a hinge to the P-domain is much too far away from the Asp[369] to allow the transfer of ATP's terminal phosphate to its aspartyl-phosphorylation site. In order to get information for how the transfer of the $\gamma$-phosphate group of ATP to the Asp[369] is achieved, analogous molecular modeling of the $M_4$–$M_5$ loop of ATPase was performed using the crystal data of Na+/K+-ATPase of different species. Analogous molecular modeling of the cytoplasmic loop between Thr[338] and Ile[760] of the $\alpha_2$-subunit of Na+/K+-ATPase and the analysis of distances between the ATP binding site and phosphorylation site revealed the existence of two ATP binding sites in the open conformation; the first one close to Phe[475] in the N-domain, the other one close to Asp[369] in the P-domain. However, binding of $Mg^{2+}\bullet$ATP to any of these sites in the "open conformation" may not lead to phosphorylation of Asp[369]. Additional conformations of the cytoplasmic loop were found wobbling between "open conformation" <==> "semi-open conformation <==> "closed conformation" in the absence of $2Mg^{2+}\bullet$ATP. The cytoplasmic loop's conformational change to the "semi-open conformation"—characterized by a hydrogen bond between Arg[543] and Asp[611]—triggers by binding of $2Mg^{2+}\bullet$ATP to a single ATP site and conversion to the "closed conformation" the phosphorylation of Asp[369] in the P-domain, and hence the start of Na+/K+-activated ATP hydrolysis.

## INTRODUCTION

$Na^+/K^+$-ATPase (EC 3.6.3.9) is an integral membrane protein that transports sodium and potassium ions against an electrochemical gradient. It belongs to the family of P-type ATPases that is structurally typified by the L-2-haloacid dehalogenase. $Na^+/K^+$-ATPase and $Ca^{2+}$-ATPase belong to this family and show a high degree of homology, especially at the phosphorylation domain. The tertiary structure of $Na^+/K^+$-ATPase has been solved at high resolution by X-ray crystallography (*Kanai et al., 2013*; *Laursen et al., 2015*; *Laursen et al., 2013*; *Morth et al., 2011*; *Morth et al., 2007*; *Nyblom et al., 2013*; *Ogawa et al., 2015*; *Ogawa et al., 2009*; *Shinoda et al., 2009*; *Yatime et al., 2011*) and also partially several N-domain structures by X-ray crystallography (*Håkansson, 2003*) and NMR (*Mark Hilge et al., 2003*). In addition, several crystallographic structures of $Ca^{2+}$-ATPase were reported (*Akin et al., 2013*; *Bublitz et al., 2015*; *Clausen et al., 2013*; *Drachmann et al., 2014*; *Jensen et al., 2006*; *Laursen et al., 2009*; *MacLennan & Green, 2000*; *Moncoq, Trieber & Young, 2007*; *Obara et al., 2005*; *Olesen et al., 2007*; *Olesen et al., 2004*; *Paulsen et al., 2013*; *Sacchetto et al., 2012*; *Sohoel et al., 2006*; *Sonntag et al., 2011*; *Sorensen, Moller & Nissen, 2004*; *Takahashi, Kondou & Toyoshima, 2007*; *Toyoshima, 2008*; *Toyoshima et al., 2013*; *Toyoshima & Mizutani, 2004*; *Toyoshima et al., 2000*; *Toyoshima & Nomura, 2002*; *Winther et al., 2010*; *Winther et al., 2013*).

The $Na^+/K^+$-ATPase consists of three subunits, the catalytic $\alpha$-subunit with a molecular mass of about 110 kDa, the $\beta$-subunit, a glycoprotein with the molecular mass of 40–60 kDa (neglecting the oligosaccharides) and eventually the associated $\gamma$-subunit with the molecular mass of 8–14 kDa (*Collins & Leszyk, 1987*; *Forbush, Kaplan & Hoffman, 1978*). The $\alpha$-subunit carries out all ion transport and catalytic functions. The ion transport of $Na^+$ and $K^+$ catalyzed by $Na^+/K^+$-ATPase in this subunit is believed to occur via transition between two major conformational states, the $E_1Na^+$ and the $E_2K^+$-conformations (*Kaplan, 2002*). The $\alpha$-subunit contains in a large cytoplasmic loop between the $M_4$ and $M_5$ transmembrane helices the catalytic center binding and hydrolyzing ATP. This large loop protruding far to cytoplasm comprises quite rigid subdomains and self-supporting substructures (*Amler, Abbott & Ball, 1992*). Structurally it consists of two main parts, the rigid nucleotide binding domain (N-domain) (*Kanai et al., 2013*; *Nyblom et al., 2013*) roughly between the amino acid residues $Arg^{380}$–$Arg^{583}$, and the domain forming the $Asp^{369}$-phosphointermediate during ATP hydrolysis (P-domain).

The secondary structure of N-domain shows a seven-stranded antiparallel $\beta$-sheet with two helix bundles sandwiching it. In this domain $Phe^{548}$, $Glu^{505}$, $Lys^{501}$, $Gln^{482}$, $Lys^{480}$, $Ser^{477}$, $Phe^{475}$ and $Glu^{446}$ participate in docking of the $Mg^{2+}$ATP complex into its binding pocket (*Kubala et al., 2003*).

The P-domain consists of two parts (subdomains). Its N-terminal subdomain ranges from $Lys^{347}$ to the residue of phosphorylation $Asp^{369}$. It is connected to the fourth transmembrane segment $M_4$ of the $\alpha$-subunit. A highly negatively charged surface was found around the phosphorylation site accessible by the solvent (*Tejral et al., 2007*; *Tejral et al., 2009*). The C-terminal subdomain formed by $Ala^{590}$–$Phe^{747}$ is connected to the fifth transmembrane segment $M_5$. These two parts (subdomains) form a typical Rossmann

fold. The secondary structure of this domain can be divided into a seven-stranded parallel $\beta$-sheet with eight short associated helices (*Morth et al., 2007*; *Ogawa et al., 2009*; *Shinoda et al., 2009*).

Despite the relatively large amount of information available on the 3-D structure of Na$^+$/K$^+$-ATPase, the molecular mechanism of the transphosphorylation process of the terminal $\gamma$-phosphate group of ATP residing in the N-domain to the Asp$^{369}$-acceptor group at the P-domain is still a puzzle. Evidently, the N-domain must bend to the P-domain by way of a mobile hinge structure. It is not clear, however, how this process is achieved on a molecular level. Hence, we tried to get information on this question using molecular modeling.

## METHODS

### Comparative modeling of the open conformation

As the solved crystal structures of Na$^+$/K$^+$-ATPase are only of a non-human origin (*Håkansson, 2003*; *Kanai et al., 2013*; *Laursen et al., 2015*; *Laursen et al., 2013*; *Mark Hilge et al., 2003*; *Morth et al., 2011*; *Morth et al., 2007*; *Nyblom et al., 2013*; *Ogawa et al., 2015*; *Ogawa et al., 2009*; *Shinoda et al., 2009*; *Yatime et al., 2011*), we decided to employ the procedure of homology modeling to get its human $\alpha_2$-subunit 3-D-structure. The primary amino acid sequence of the human Na$^+$/K$^+$-ATPase was retrieved from the ExPASy server (UniProt KB/TrEMBL; http://www.expasy.ch/). The resulting P50993 (AT1A2_HUMAN) target human sequence in the length of 1,020 amino acids for the Na$^+$/K$^+$-ATPase $\alpha_2$-subunit precursor of sodium/potassium-transporting ATPase $\alpha_2$-subunit, Homo sapiens, EC 3.6.3.9, was chosen (*Shull, Pugh & Lingrel, 1989*). Five amino acids at the N-terminal beginning of this sequence compared to the translated RNA sequence do not occur in the native form (*Hara et al., 1987*; *Kawakami et al., 1985*; *Ovchinnikov et al., 1986*; *Shull, Schwartz & Lingrel, 1985*). Hence, they were not included in our further numbering. For modeling, known structures of Na$^+$/K$^+$-ATPase deposited at the RCSB Protein Data Bank (http://www.pdb.org/) were used. In order to create the model based on the above mentioned sequence, the solved crystal structures of Na$^+$/K$^+$-ATPase with RCSB Protein Data Bank (http://www.pdb.org/) accession codes 3B8E (*Morth et al., 2007*) and 3KDP (*Morth et al., 2007*) were used as the templates for our modeling. The multialignment of the chosen target (P50993, AT1A2_HUMAN) sequence and the two templates (3B8E, 3KDP) for open conformation was prepared by MODELLER program (salign module) (*Eswar et al., 2006*; *Marti-Renom et al., 2000*; *Šali, 1995*; *Šali & Blundell, 1993*). The choice of templates (solved crystal structures) was based on the species proximity (pig over shark) and absence of any cardioglycosides in the solved crystal structure, in order to get as close to the native form as possible. Using this multialignment and the solved 3D crystal structures, we have generated thousand M$_4$M$_5$-loop models by the MODELLER (automodel module) (*Eswar et al., 2006*; *Marti-Renom et al., 2000*; *Šali, 1995*; *Šali & Blundell, 1993*) program. From those created models the best thirty were selected using the PROCHECK (*Laskowski et al., 1993*; *Morris et al., 1992*) and Verifi3D (*Bowie, Luthy & Eisenberg, 1991*; *Lüthy, Bowie & Eisenberg, 1992*) programs (Table 1).

**Table 1** The assessment of homology model quality (compared to the ones of template crystal structures).

| Model | Compound (pdb code) | Ramachandran plot: percent of aminoacids in allowed regions | Procheck: overall G-factor | Verify3D: percent of residues that had an averaged 3D-1D score $\geq$ 0.2 | Total energy (kJ/mol) (GROMOS96) |
|---|---|---|---|---|---|
| Open conformation | 3b8eA (crystal structure) | 93.6 | 0.01 | 88.44 | −4645.14 |
| | 3b8eC (crystal structure) | 93.3 | 0.01 | 91.98 | −4644.72 |
| | 3kdpA (crystal structure) | 95.8 | −0.31 | 92.69 | −3484.95 |
| | 3kdpC (crystal structure) | 96.3 | −0.32 | 92.69 | −2921.80 |
| | Model of open conformation | 98.2 | −0.21 | 97.64 | −14113.98 |
| Closed conformation | 3wguA (crystal structure) | 98.9 | −0.07 | 99.53 | −10732.51 |
| | 3wguC (crystal structure) | 99.8 | 0.01 | 97.17 | −13461.95 |
| | 3wgvA (crystal structure) | 98.7 | −0.10 | 96.70 | −10132.11 |
| | 3wgvC (crystal structure) | 100.0 | −0.02 | 95.99 | −12870.88 |
| | 4hqjA (crystal structure) | 99.2 | 0.10 | 98.58 | −12983.13 |
| | 4hqjC (crystal structure) | 99.4 | 0.13 | 98.58 | −13081.49 |
| | Model of closed conformation | 99.5 | 0.06 | 96.93 | −7995.71 |

## Comparative modeling of the closed conformation

As in the previous comparative modeling procedure, we have used the sequence P50993 (AT1A2_HUMAN) for modeling of $Na^+/K^+$-ATPase in the closed conformation. However, the solved crystal structures of the RCSB Protein Data Bank (http://www.pdb.org/—accession codes 3WGU, 3WGV and 4HQJ (*Kanai et al., 2013*; *Nyblom et al., 2013*)) were used as the templates for our modeling. Using the above-mentioned settings for the modeling program (MODELLER see previous paragraph) we obtained ten models of $Na^+/K^+$-ATPase in the closed conformation. From these, the best model has been chosen using the above-mentioned PROCHEK and Verifi3D programs (Table 1).

## Docking, using the open and closed conformations

The best thirty models corresponding to the open conformation and the best model for the closed conformation were used for docking of $Mg^{2+}$•ATP complex, using the Vina-Autodock program (*Trott & Olson, 2010*). We have decided to use the whole $Mg^{2+}$•ATP complex, which has been derived from structures containing ATP, deposited in RCSB Protein Data Bank (http://www.pdb.org/). The sequential docking of $Mg^{2+}$, followed by ATP, has not been used, since the bond between $Mg^{2+}$ and ATP phosphates is stronger than between $Mg^{2+}$ and –COOH groups of amino acids (*Alberty, 1969*; *Dudev, Cowan & Lim, 1999*). In addition, the same procedure was used for the model of the closed conformation.

## Molecular dynamics

Molecular dynamics (MD) of the $M_4M_5$-loop were simulated by Gromacs (*Berendsen, Van der Spoel & Van Drunen, 1995*; *Hess et al., 2008*; *Lindahl, Hess & Van der Spoel, 2001*; *Van der Spoel et al., 2005*; *Van der Spoel et al., 2010*), using the OPLS-AA potential (*Jorgensen, Maxwell & Tirado-Rives, 1996*; *Jorgensen & Tirado-Rives, 1988*; *Kaminski et al., 2001*; *Meagher, Redman & Carlson, 2003*; *Pranata, Wierschke & Jorgensen, 1991*) with combination of water model TIP3P (*Jorgensen et al., 1983*). Our protein model with or without

$2Mg^{2+}$•ATP were put into a rectangular box with a 1 nm thick layer of the water molecules around and periodic boundary conditions (*Berendsen, Van der Spoel & Van Drunen, 1995*; *Hess et al., 2008*; *Lindahl, Hess & Van der Spoel, 2001*; *Van der Spoel et al., 2005*; *Van der Spoel et al., 2010*).

The PME method (*Darden, York & Pedersen, 1993*) with a length parameter of 1 nm was used to describe Coulomb type electrostatic interactions and the cut-off method with a length parameter of 1 nm for the calculation of van der Waals interactions. As the first step of the MD simulation, the system of protein and water was energetically optimized using the method of steepest descents, followed by a conjugate gradient minimization algorithm with maximum $2.5 \times 10^4$ steps and maximum force smaller than 10 kj mol$^{-1}$ nm$^{-1}$ as the convergence criterion (see Supplemental Information 1). The *Berendsen et al. (1984)* coupling method was employed for the temperature and pressure coupling of a system to reflect the reference temperature of 310 K and the pressure of 1 bar. The leap-frog integration with $10^4$ steps was used for stabilization, with integration step of 1 fs, corresponding to 10 ps simulation time to reach the equilibrium of the rectangular box (see Supplemental Information 1). This stabilized rectangular box was used for the main thirty simulations with $5 \times 10^6$ steps (2 fs single step), corresponding to 10 ns for each stabilization using the same simulation parameters as for the box stabilization. These thirty trajectories were simulated with independently generated initial conditions corresponding to a Maxwell distribution for a temperature of 310 K. The translation and rotation around the center of mass of the protein were removed, avoiding thus the simulated system distortion in the simulation box. The molecular dynamics simulation of the Na$^+$/K$^+$-ATPase in semi-open conformation with docked $2Mg^{2+}$•ATP was carried out using the same parameters as described above.

### Docking to the semi-open conformation

From the charts of the time evolution of distances, the typical trajectory for the molecular dynamics of the open conformation without $2Mg^{2+}$•ATP has been chosen. The model showing a distance between Asp$^{369}$ and Phe$^{475}$ smaller than 2.2 nm and with the best PROCHECK and Verify3D scores were taken for docking of the $2Mg^{2+}$•ATP complex.

## RESULTS

### Assembly of the static 3D computational model of Na$^+$/K$^+$-ATPase

The main goal of this work was to describe the molecular mechanism of $\gamma$-phosphate transfer from the ATP in the binding site to the phosphorylation site (Asp$^{369}$) of Na$^+$/K$^+$-ATPase. To achieve this, first, a static three-dimensional model of Na$^+$/K$^+$-ATPase was developed based on the latest data and information. The P50993 (AT1A2_HUMAN) human target sequence of the Na$^+$/K$^+$-ATPase $\alpha_2$-isoform (sodium/potassium-transporting ATPase $\alpha_2$-subunit, Homo sapiens) was used for modeling and 3D model assembling. Two templates of Na$^+$/K$^+$-ATPase structures of accession codes 3B8E (*Morth et al., 2007*) and 3KDP (*Morth et al., 2007*) were retrieved from the Protein Data Bank which were proposed by UniProt server as sequence P05024 (sodium/potassium-transporting ATPase $\alpha_1$-subunit, Sus scrofa) with sequences identity 86.5% of the target sequence. The alignments for open

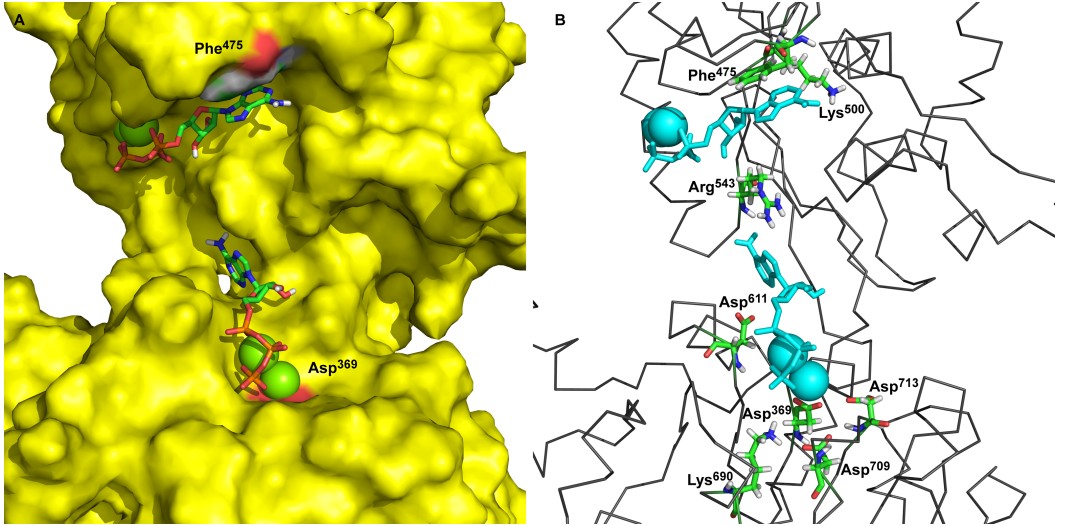

**Figure 1** **Two ATP binding sites in "open" conformation.** (A) ATP bound near Phe[475] has docking energy −7.6 kcal/mol, ATP near Asp[369] has docking energy −8.9 kcal/mol. (B) The interacting amino acids with docked $2Mg^{2+}$ATP—both binding sites.

conformation of the $M_4M_5$-loop were prepared with identity of the 85.6% between the corresponding sequences of the human $\alpha_2$-isoform and pig $\alpha_1$-isoform for the M4M5-loop. The alignment and template structures were used for comparative modeling using the MODELLER program. The obtained 3D models were verified, applying the PROCHECK and Verifi3D programs (Table 1). Our modeling procedure resulted in static structures of the $M_4M_5$-loop of human $\alpha_2$ isoform of $Na^+/K^+$-ATPase between Thr[338] and Ile[760] (see Fig. 1, with docked $2Mg^{2+}$•ATP). These models show distances around 3.26 nm between Phe[475] as part of the ATP-binding site and the $\alpha$-carbon of Asp[369], the acceptor site for the phosphointermediate in ATP hydrolysis.

## Two ATP binding sites exist in the open conformation of $Na^+/K^+$-ATPase

The obtained models of the open conformation of $Na^+/K^+$-ATPase were tested by an *in silico* ATP-docking experiment for its ability to bind $Mg^{2+}$•ATP. Surprisingly, we identified two possible docking sites (Fig. 1): the first one is in closest vicinity to Phe[475] ("the Phe[475] location") and the second one is close to Asp[369], ("the Asp[369] location"). Both binding sites showed only slightly different docking energies. While the docking energy at the Phe[475] location was $E_b = -7.6$ kcal/mol, the docking energy at the Asp[369] location was $E_b = -8.6$ kcal/mol. A closer insight into our model clearly indicated interactions among $\pi$-electrons between Phe[475] and the ATP adenine ring at the Phe[475] location, but the interaction between ATP's phosphates with bound magnesium and the negatively charged aspartate residue was responsible for the $2Mg^{2+}$•ATP binding at the Asp[369] location.

The amino acids found in the neighborhood of docked $Mg^{2+}$•ATP in both binding sites were identified and found to be in agreement with already published data (*Jacobsen, Pedersen & Jorgensen, 2002*; *Jorgensen, Hakansson & Karlish, 2003*; *Jorgensen, Jorgensen*

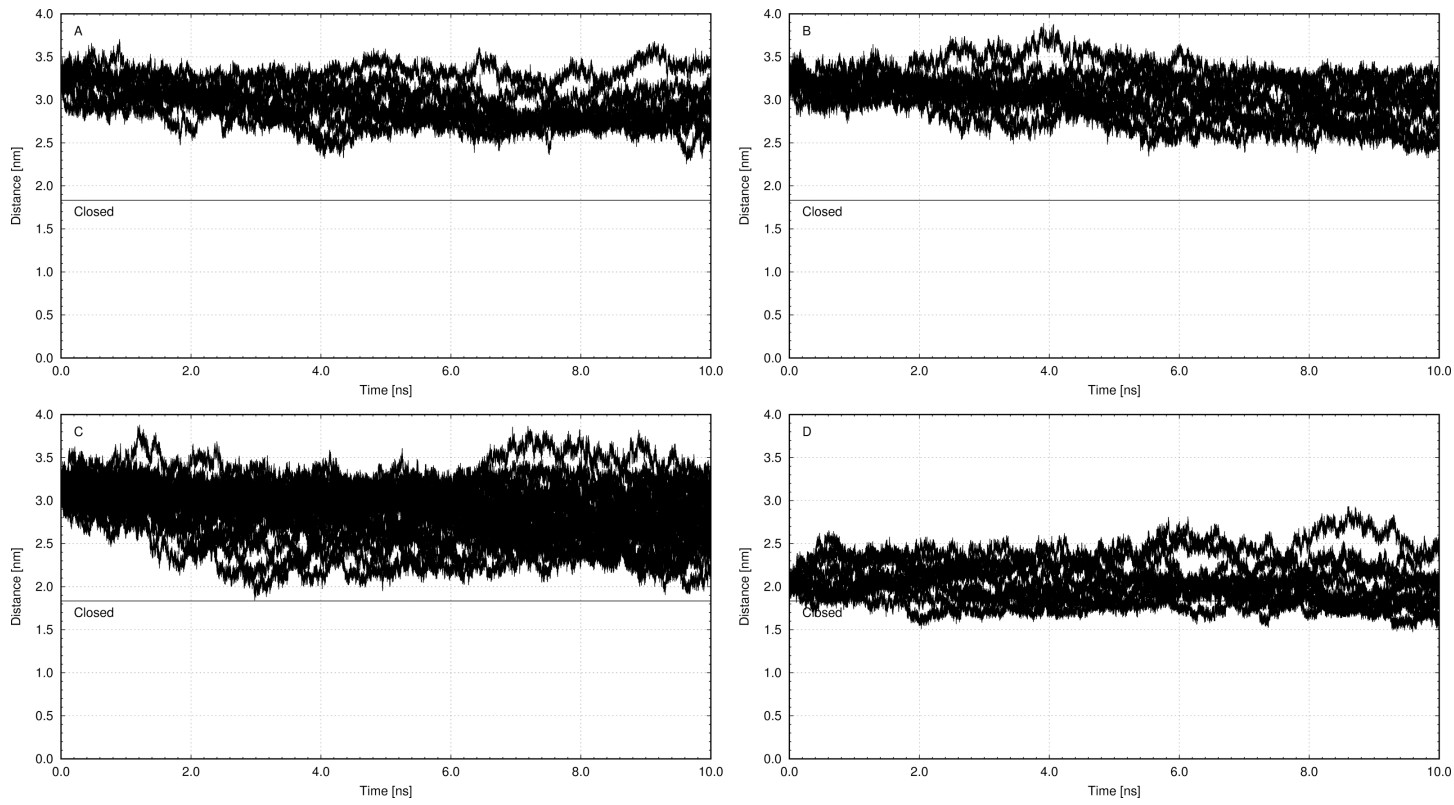

**Figure 2** (A) Molecular dynamics simulation of the model in open state (change of distance between Phe[475] and Asp[369] during simulation) with $2Mg^{2+}$ ATP interacting with Phe[475]. (B) Molecular dynamics simulation of the model in open state (change of distance between Phe[475] and Asp[369] during simulation) $2Mg^{2+}$ ATP interacting with Asp[369]. (C) Molecular dynamics simulation of the model in open state (change of distance between Phe[475] and Asp[369] during simulation) without $2Mg^{2+}$ ATP. (D) Molecular dynamics simulation of the model in open state (change of distance between Phe[475] and Asp[369] during simulation) with $2Mg^{2+}$ ATP docked in semi-open conformation (C), and interacting with both Phe[475] and Asp[369] (closed conformation).

*& Pedersen, 2001*; *Jorgensen & Pedersen, 2001*; *Kubala et al., 2003*; *Pedersen, Jorgensen & Jorgensen, 2000*).

## High ATP concentration hinders the enzyme cycle and keeps the Na⁺/K⁺-ATPase at the open conformation

The structure with best docking energy for both $2Mg^{2+}\bullet ATP$ docking sites was the starting point for a molecular dynamics simulation, which revealed another surprising result. The molecular dynamics simulation in the presence of $2Mg^{2+}\bullet ATP$ (we ran two simulations series, one with $2Mg^{2+}\bullet ATP$ docked in the ATP binding site and the other in the phosphorylation site) did not result in a stable close conformation needed to phorphorylate Asp[369] during Na⁺/K⁺-ATP hydrolysis. Interestingly, the enzyme preferentially remained in the open conformation in both simulations as is evident from the resulting distance distribution between $\alpha$-carbons of Asp[369] and Phe[475] (Figs. 2A, 2B, 3A and 3B, the distance varied from 2.5 nm to 3.4 nm, with maxima 2.9 nm and 3.1 nm respectively).

However, molecular dynamics experiments in the absence of $2Mg^{2+}\bullet ATP$ (Fig. 3C) in nanosecond timescale exhibited a different pattern. This conformation was characterized

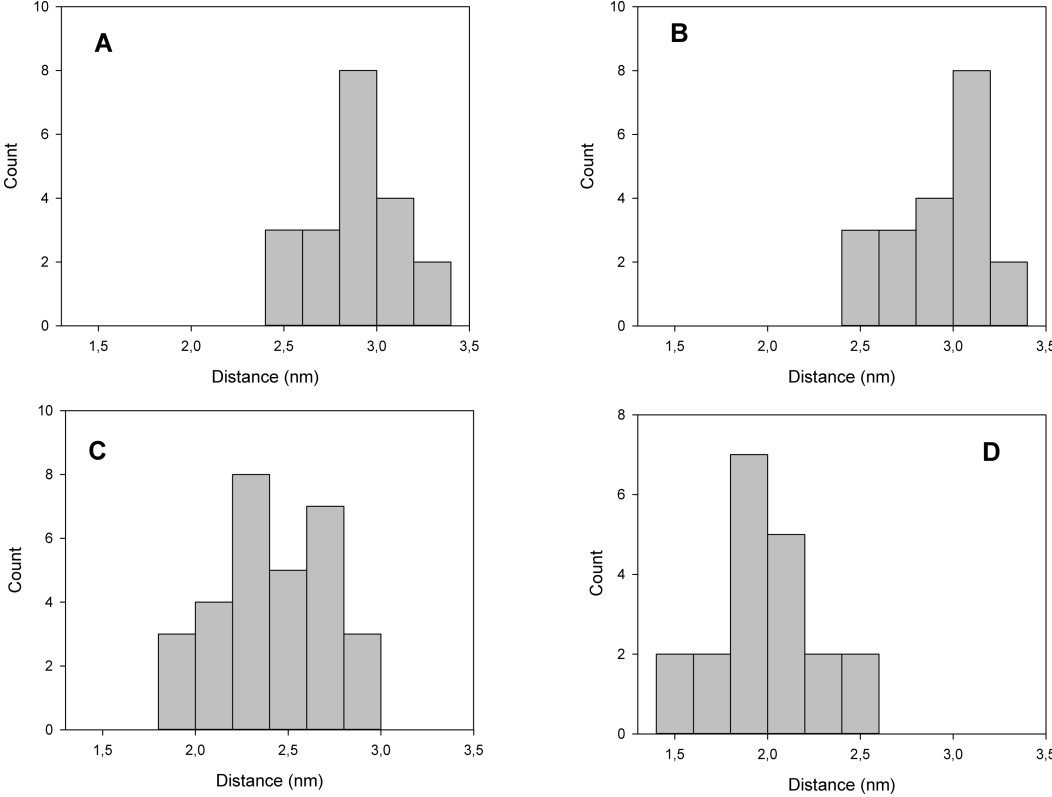

**Figure 3** (A) Resulting distance distribution between Phe[475] and Asp[369] at the end of simulation with $2Mg^{2+}ATP$ interacting with Phe[475]. (B) Resulting distance distribution between Phe[475] and Asp[369] at the end of simulation with $2Mg^{2+}ATP$ interacting with Asp[369]. (C) Resulting distance distribution between Phe[475] and Asp[369] at the end of simulation without $2Mg^{2+}ATP$. (D) Resulting distance distribution between Phe[475] and Asp[369] at the end of simulation with $2Mg^{2+}ATP$ docked in semi-open conformation (C), and interacting with both Phe[475] and Asp[369] (closed conformation).

by shortening of the distance between $\alpha$-carbons of Asp[369] and Phe[475] to about $d \sim 2.00$ nm (Fig. 3C).

Additionally, we performed 30 simulations in the absence of $2Mg^{2+} \bullet ATP$. Yet, there was no stable result: sometimes, the molecular dynamic simulation led to the new conformation (we will call this conformation "semi-open" conformation), but sometimes the enzyme remained in the open conformation, with the ratio open/semi-open conformation being approximately 1:1.4 (Figs. 2C and 3C). Clearly, conformational transitions between the "open" and "semi-open conformations" seem to be rather a stochastic process (Figs. 2C and 3C). Consequently, we decided to call this newly identified conformation representing a distance of 2.3 nm between Phe[475] and Asp[369] the "semi-open" conformation of the catalytic site of $Na^+/K^+$-ATPase. The semi-open conformation is characterized by the formation of a hydrogen bond between Arg[543] and Asp[611] (see Fig. 4).
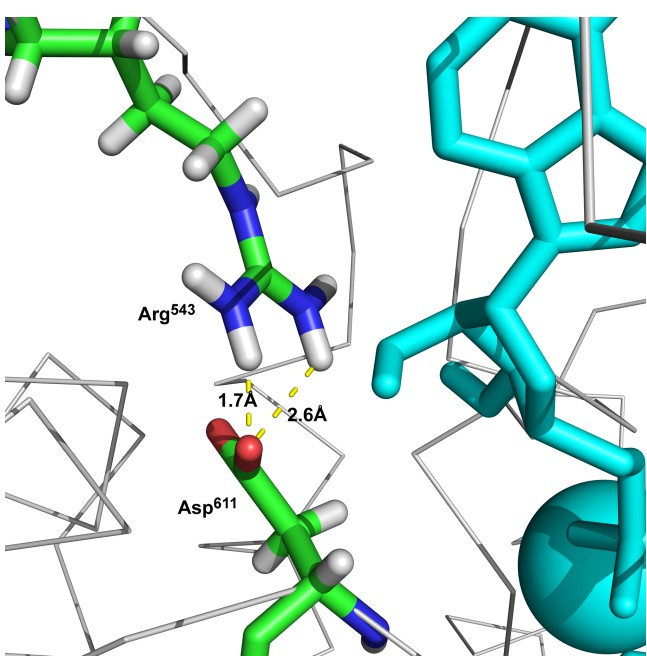

**Figure 4** The hydrogen bonds between Arg[543] and Asp[611] formed in semi-open state.

## Hinge movement, ATP binding and enzyme phosphorylation

Release and re-binding of $2Mg^{2+} \bullet ATP$ complexes at two different binding sites in the open conformation may have huge consequences for the molecular mechanism of the transphosphorylation process to Asp[369] as part of the ATP hydrolysis of $Na^+/K^+$-ATPase. Most importantly is the fact that the $\alpha$-subunit of $Na^+/K^+$-ATPase can wobble between the "open" and "semi-open conformations" in the absence of $2Mg^{2+} \bullet ATP$.

Naturally, the obvious question arises, whether and how the $2Mg^{2+} \bullet ATP$ complex interacts with the "semi-open conformation." Therefore, $2Mg^{2+} \bullet ATP$ molecule has been docked into the "semi-open conformation" (Fig. 5), revealing only a single $2Mg^{2+} \bullet ATP$ binding site exists. This $2Mg^{2+} \bullet ATP$ binding site in the "semi-open conformation" was formed as a sandwich structure from both, "the Phe[475] location" and "the Asp[369] location" as they were revealed and identified at the open conformation. Both sites have approached each other due the stochastic process in the absence of $2Mg^{2+} \bullet ATP$, probably due to the preceding hinge movement in the absence of $2Mg^{2+} \bullet ATP$. This binding pocket for a single $2Mg^{2+} \bullet ATP$ is characterized by the most favorable and highest docking energy of $E_b = -8.8$ kcal/mol.

Furthermore, docking of $2Mg^{2+} \bullet ATP$ to the "semi-open conformation" results in a further substantial shortening of the distance between the Asp[369] and Phe[475]. Consequently, the $\gamma$-phosphate of $2Mg^{2+} \bullet ATP$ was attracted to Asp[369] and the mutual distance between the $\alpha$-carbons of Asp[369] and Phe[475] decreased to approximately $d = 1.8$ nm (Figs. 2D and 3D, majority falling in the interval 1.5–2.0 nm). This shortening can be explained as a consequence of the second phase of the hinge movement: bending of the N-domain toward the P-domain, which completes "the hinge mechanism" (Fig. 6).
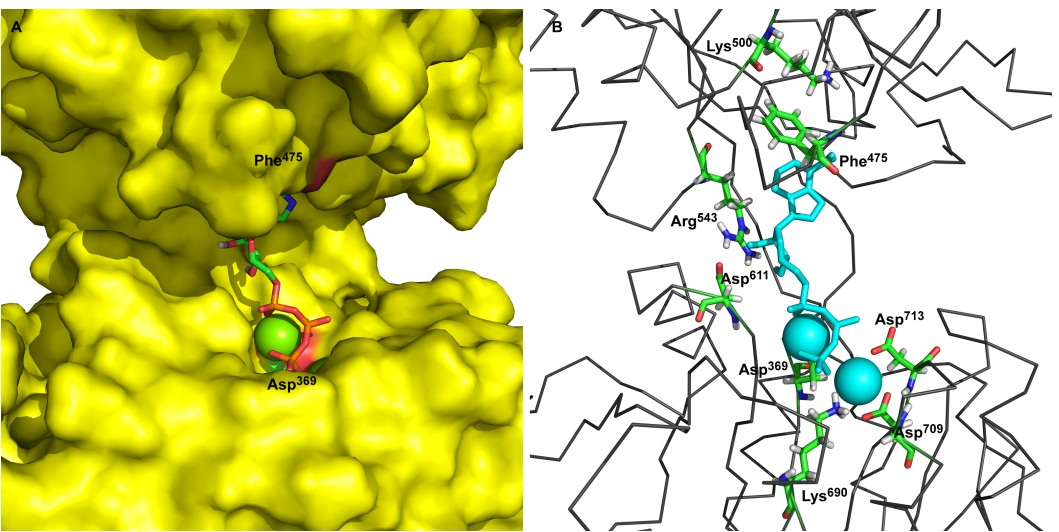

**Figure 5** **Docking of 2Mg$^{2+}$•ATP to the "semi-open" conformation.** The simultaneous interaction of 2Mg$^{2+}$•ATP with Phe$^{475}$ and Asp$^{369}$ can be identified.

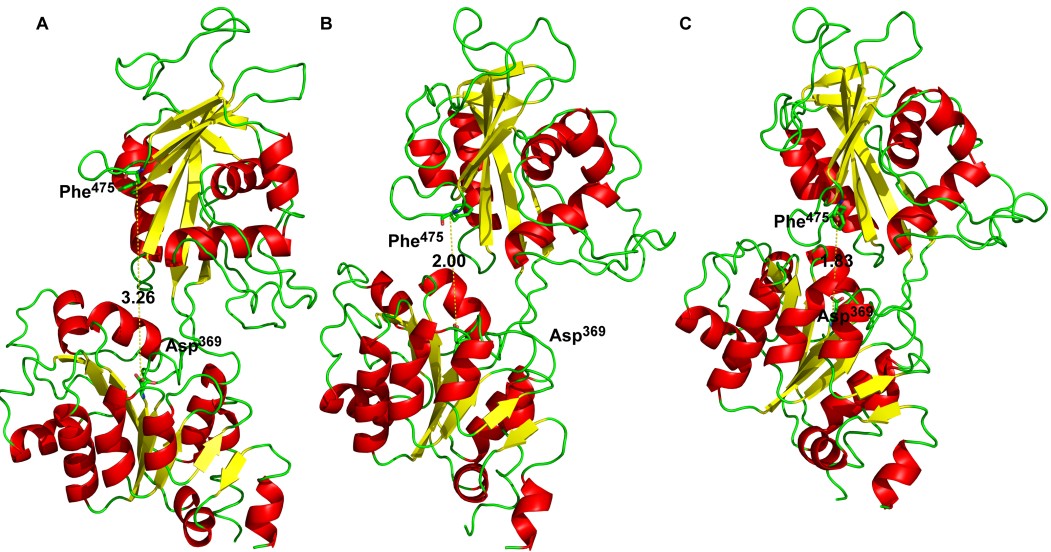

**Figure 6** **The three conformational states of Na$^+$/K$^+$-ATPase (distances are in Å).** From the left: (A) Open conformation, (B) "Semi-open" conformation and (C) Closed conformation.

In order to verify our conclusions, the closed conformation of Na$^+$/K$^+$-ATPase structure was prepared by homology modeling as well using crystallography templates (Fig. 7). The docking experiment of 2Mg$^{2+}$•ATP to the "closed" conformation, revealed the existence of a single ATP binding site as well (Fig. 7). Moreover, the molecular dynamic experiment with the "semi-open" sub-conformation shows that it's conformation differs from that one of the "closed" conformation with overall RMSD < 0.3 nm, which is within the experimental error of crystallographic data.

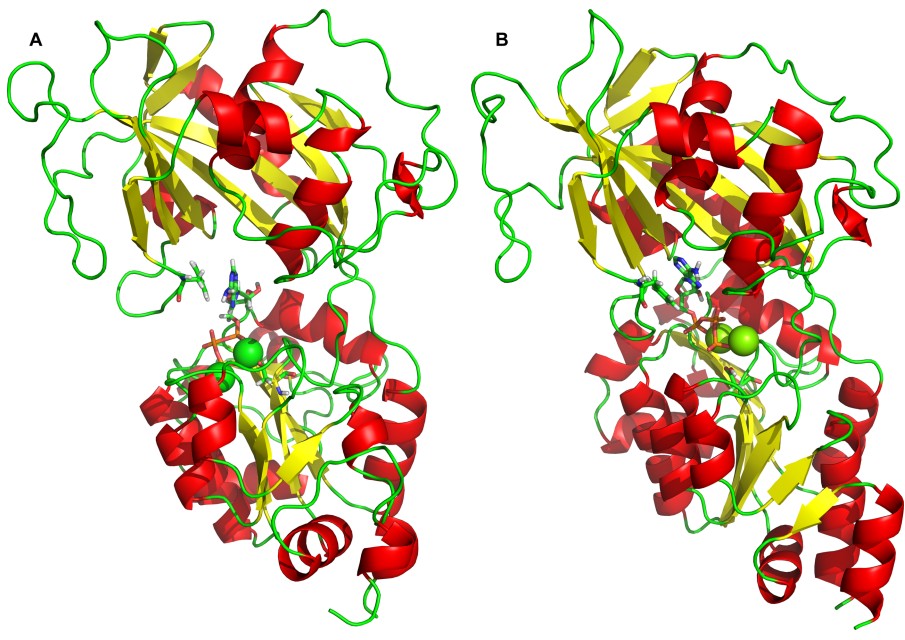

**Figure 7** (A) The conformation received as the result of the molecular dynamic experiment with docked $2Mg^{2+} \bullet ATP$ (closed state). (B) The conformation received as the result of the homology modeling of $Na^+/K^+$-ATPase in the "closed" state, with docked $2Mg^{2+} ATP$. The overall difference between these two conformations has RMSD = 0.27 nm, which is within the error of the crystallographic data.

## DISCUSSION

### Multiple ATP binding sites are found in the open and semi-open conformations of the cytoplasmic $M_4$–$M_5$-loop of $Na^+/K^+$-ATPase

The intention of this work was to learn by inspection of a large number of related and crystallized P-type ATPases and analogous computer modeling of the cytoplasmic $M_4$–$M_5$-loop of the human $\alpha_2$ isoform of $Na^+/K^+$-ATPase, how on a molecular level the distance is shortened between the nucleotide binding site (the N-domain) and the phosphorylation site $Asp^{369}$ at the P-domain. The distance of 3.26 nm between both sites (in the "open" state, Fig. 6) is too high to support either the $Na^+ + Mg^{2+}$ or the $Mg^{2+}$-dependent transphosphorylation process or the ATP–ADP exchange reaction (*Fahn, Koval & Albers, 1966*), both the partial reactions of $Na^+/K^+$-ATPase. Evidently, any changes by binding of $Na^+$ or $K^+$ to their respective membrane sites must be excluded, since our analysis was restricted exclusively to the molecular events at the large cytoplasmic $M_4$–$M_5$-loop: We intended to understand the bending mechanism of the N-domain towards the P-domain.

### ATP binding into the semi-open conformation leads to the hinge movement and triggers enzyme phosphorylation

We identified by molecular modeling of the cytoplasmic loop structure the existence of 3 different conformational states with the ability to bind ATP (Fig. 6). In the absence of ATP and $2Mg^{2+} \bullet ATP$ the "open" and "semi-open" conformational states are freely interconverted. The open state binds ATP in the absence of $Mg^{2+}$ to the N-domain as previously shown (References). It may bind, however, also $2Mg2^+ \bullet ATP$ at 2 sites, the "the $Phe^{475}$

PeerJ ________________________________________________

location" and the "the Asp[369] location" (Fig. 1). Yet, in this open conformation, no transphosphorylation of the gamma phosphate group of ATP to Asp[369] residing on the P-domain is possible: the terminal phosphate of ATP is much too remote from the carboxyl group of Asp[369]. Molecular modeling clearly showed that it is rather the newly identified semi-open conformation which binds $2Mg^{2+} \bullet ATP$ to a single site in such a way that the terminal phosphate approaches the phosphate acceptor site Asp[369] on the P-domain leads and via a further shift to the "occluded" state may achieve its phosphorylation.

On a molecular level our model describes and is in agreement with the vast majority of published structures for the ATP binding domains of P-type ATPases (*Bublitz et al., 2013*; *Bueno-Orovio et al., 2014*; *Castillo et al., 2011*; *Castillo et al., 2015*; *Fuller et al., 2013*; *Howland, 1991*; *Jacobsen, Pedersen & Jorgensen, 2002*; *Jensen et al., 2006*; *Jorgensen, Hakansson & Karlish, 2003*; *Jorgensen, Jorgensen & Pedersen, 2001*; *Jorgensen & Pedersen, 2001*; *Kanai et al., 2013*; *Kaplan, 2002*; *Laursen et al., 2015*; *MacLennan & Green, 2000*; *Morth et al., 2007*; *Nyblom et al., 2013*; *Obara et al., 2005*; *Ogawa et al., 2009*; *Olesen et al., 2007*; *Pedersen, Jorgensen & Jorgensen, 2000*; *Sacchetto et al., 2012*; *Shinoda et al., 2009*; *Toyoshima, 2008*; *Toyoshima et al., 2013*; *Toyoshima & Mizutani, 2004*; *Toyoshima et al., 2000*). Importantly, Arg[543] is located in the N domain near the interface to the P domain (Fig. 1). This residue has been shown to be essential for nucleotide binding; its substitution by Gln abolishes high-affinity binding of ATP (in the absence of $Mg^{2+}$) and also $Na^+/K^+$-ATPase activity (*Pedersen, Jorgensen & Jorgensen, 2000*). The free energy required overcoming the electrostatic interactions between the $\gamma$-phosphate of $2Mg^{2+} \bullet ATP$ and the carboxylate groups amounts to 7.9 kcal/mol for Asp[369]. This value supports our model exactly. In addition, the increased binding energy of $2Mg^{2+} \bullet ATP$ is connected with a conformational transition constituting the driving force for transport of $K^+$ across the membrane (*Howland, 1991*). Additionally, our molecular modeling experiments showed in docking experiments a very favorable binding energy of $2Mg^{2+} ATP$ at the semi-open conformation. The strong electrostatic interaction with the negative charges of Asp[369], Asp[709] and Asp[713] with $2Mg^{2+} \bullet ATP$ shows that the $\gamma$-phosphate of the tightly bound ATP are important to approach the surface of the P domain in $Na^+/K^+$-ATPase (*Jorgensen, Hakansson & Karlish, 2003*; *Jorgensen, Jorgensen & Pedersen, 2001*; *Jorgensen & Pedersen, 2001*). This certainly leads to further bridging the gap between the N- and P-domains and the formation of a "closed conformation" (Fig. 6C) resulting in a type of "occluded $2Mg^{2+} \bullet ATP$" preceding the formation of a phosphointermediate in the ATP-$E_1$ form of the $\alpha$ subunit of $Na^+/K^+$-ATPase.

Analogously, in the crystal structure of $Ca^{2+}$-ATPase in the $E_1[2Ca^{2+}$-] form (*Clausen et al., 2013*; *Toyoshima et al., 2013*; *Winther et al., 2013*), the N domain is separated from the P domain by a distance of 2.0–2.5 nm. Such a distance is also seen in our model in the "open" conformational state: the P domain of the human $\alpha_2$ isoform of $Na^+/K^+$-ATPase is separated from the N domain by a distance of less than 2 nm. Additionally, Lys[690] appears to create a salt linkage with the phosphate group as has been found in previous experiments. $Mg^{2+}$ is essential for all phosphoryl transfer reactions. The experience from $Mg^{2+}$ binding studies is that the binding affinity and the coordination pattern depend strongly on the conformational state (*Pedersen, Jorgensen & Jorgensen, 2000*) . Our model shows this as well. Importantly, fluctuation in between the "open" and "semi-open

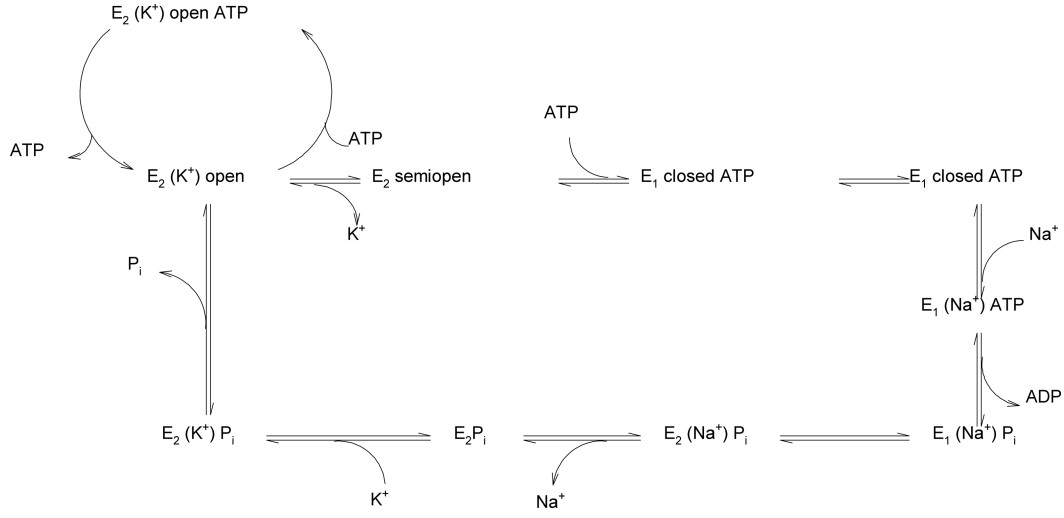

**Figure 8** Reaction scheme of Na⁺/K⁺-ATPase explaining the possible functions of the "open", "semi-open" and "closed" conformations of the big cytoplasmic loop within the export of Na⁺ and import of K⁺ by the sodium pump.

conformations" is connected with binding/unbinding of $2Mg^{2+}\bullet ATP$ to the three above mentioned negatively charged residues of $Asp^{709}$, $Asp^{713}$ and $Asp^{369}$ (Figs. 1B and 4). However, binding of $2Mg^{2+}\bullet ATP$ to any site of the "open conformation" cannot lead to phosphorylation. The phosphorylation process can be triggered only when the "semi-open conformation" in the absence of ATP is formed. Once the semi-open conformation has been created, by forming a hydrogen bond between $Arg^{543}$ and $Asp^{611}$ (Fig. 4), the affinity for ATP peaks, facilitating thus ATP binding. The distance between $Phe^{475}$ and $Asp^{369}$ decreased to about 1.8 nm (corresponding to 1.83 in our "closed conformation" model). This average value of the experimentally reported distances for the "closed conformation", enables the phosphorylation of $Asp^{369}$, and is in accordance with the measured data (*Jacobsen, Pedersen & Jorgensen, 2002*; *Jorgensen, Jorgensen & Pedersen, 2001*).

To the best of our knowledge, this is the first report on the existence of three conformers of the big cytoplasmic loop binding ATP. Our finding may have important consequences for understanding the molecular mechanism of the Na⁺/K⁺-ATPase function. Na⁺ ions have been reported to increase the activity of transphosphorylation process (*Kaplan, 2002*). It is unclear at present, where Na⁺ binds to its transport site in the transmembranal part of the enzyme and how this may affect the conformational transitions of the hinge region in the closing process approaching N- and P-domains such a way that the phosporylation of $Asp^{369}$ as an intermediate may happen. It needs to be investigated in further studies, how, at a molecular level, high ATP concentrations lead to the release of $E_2$-occluded K⁺. Micromolar ATP concentrations are sufficient for ATP binding (in the absence of $Mg^{2+}$) to the N-domain in the open state (*Kubala et al., 2003*; *Schoner, Beusch & Kramer, 1968*; *Tran & Farley, 1999*). The effect of $Mg^{2+}$ on the binding of ATP to the isolated N-domain has never been studied. Micromolar ATP concentrations are sufficient for the Na⁺ + $Mg^{2+}$-dependent formation of the $Asp^{369}$-phosphointermediate (*Hegyvary & Post, 1971*;

*Moczydlowski & Fortes, 1981*). Millimolar ATP concentrations are necessary for the overall $Na^+/K^+$-activated ATP hydrolysis necessary for $Na^+/K^+$-transport. High (millimolar) ATP concentrations are necessary to result in the de-occlusion of $K^+$ from its transmembrane site. Might it be that the existence of 2 ATP sites in the "open conformation" of the cytoplasmic loop (Fig. 1) represents a situation of opening of the closed catalytic site for MgATP at high concentrations of the energy substrate. It is well known that $K^+$ ions are on its way from the outside to the inside of the cell included into the transmembrane part of $Na^+/K^+$-ATPase. High concentrations of MgATP are necessary to release occludes $K^+$ from the sodium pump into the cytoplasm. One may speculate that binding of MgATP at millimolar concentrations may lead to a shift of the "closed conformation" to the "open conformation," i.e., the displacement of the N-domain via the hinge mechanism from the P-domain due to binding of millimolar MgATP to the N-domain (Fig. 8).

### Funding
Computational resources were provided by the CESNET LM2015042 and the CERIT Scientific Cloud LM2015085, provided under the programme "Projects of Large Research, Development, and Innovations Infrastructures." This research was supported by the Czech Science Foundation Grant No. 15-15697S, the University Centre for Energy Efficient Buildings (UCEEB) support IPv6; the Ministry of Education, Youth, and Sports of the Czech Republic (National Sustainability Programme I, project No. LO1605; Research Programs NPU I:LO1508 and NPU I:LO1309); the Internal Grant Agency of the Ministry of Health of the Czech Republic (grant No. NT12156 and MZ-VES project no. 16-29680A and 16-28637A); University Hospital Motol (project 9775); the Grant Agency of Charles University (Grants No. 456216); and the Ministry of Interior of the Czech Republic (program BV III/1-VS, No VI20152018010). The funders had no role in study design, data collection and analysis, decision to publish, or preparation of the manuscript.

### Grant Disclosures
The following grant information was disclosed by the authors:
Czech Science Foundation: 15-15697S.
Ministry of Education, Youth, and Sports of the Czech Republic.
Internal Grant Agency of the Ministry of Health of the Czech Republic: NT12156.
University Hospital Motol.
Grant Agency of Charles University: 456216.
Ministry of Interior of the Czech Republic.

### Competing Interests
The authors declare there are no competing interests.

### Author Contributions
- Gracian Tejral conceived and designed the experiments, performed the experiments, wrote the paper, prepared figures and/or tables.

- Bruno Sopko conceived and designed the experiments, analyzed the data, wrote the paper, prepared figures and/or tables.
- Alois Necas wrote the paper, reviewed drafts of the paper.
- Wilhelm Schoner and Evzen Amler conceived and designed the experiments, wrote the paper, reviewed drafts of the paper.

## Data Availability

Comparative modeling of the different conformations are detailed in 'Methods'.

## Supplemental Information

Supplemental information for this article can be found online at http://dx.doi.org/10.7717/peerj.3087#supplemental-information.

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
