# Peer review of "Computer modelling reveals new conformers of the ATP binding loop of Na+/K+-ATPase involved in the transphosphorylation process of the sodium pump"

_PeerJ, doi:10.7717/peerj.3087_

## Round 0.1 · original submission · Major Revisions

Please pay close attention to the reviewers' comments and revise your manuscript accordingly.

Reviewer 1 ·

Basic reporting

The authors studied 3-dimensional molecular structure analysis of P-type
ATPases revealing that binding of ATP to the N-domain connected by a hinge to the P-domain
is much too far away from the Asp369 to allow the transfer of ATP’s terminal phosphate to
its aspartyl-phosphorylation site. In order to get information how the transfer of the
γ‑phosphate group of ATP to the Asp369 is achieved, analogous molecular modeling of the
M4-M5 loop of ATPase was performed using the crystal data of Na+/K+-ATPase of different
species. Analogous molecular modeling of the cytoplasmic loop between Thr338 and Ile760 of
the α2-subunit of Na+/K+-ATPase and the analysis of distances between the ATP binding site
and phosphorylation site revealed the existence of 2 ATP binding sites in the open
conformation, the first one close to Phe475 in the N-domain, the other one close to Asp369 in
the P-domain.

Experimental design

The design with molecular docking and molecular dynamics simulations are appropriate but thermodynamics is missing.

Validity of the findings

The findings are valid and nicely presented

Additional comments

If you could add thermodynamics to the binding using the trajectories from MD simulations utilizing the MM/PBSA method, the manuscript would be stronger.

Reviewer 2 ·

Basic reporting

I find the English style of this article somewhat difficult to read. On many occasions, departures from conventional language adopted in the field have occurred. The authors may consider taking an assistance in scientific writing specifically in this subject area.

Homology modeling used to select initial structures (lines 87-130) is not illustrated by specific scores of top models, making it impossible to understand what exactly criteria were used for the structure selection.

Experimental design

The research goals are well identified, and sound original and interesting.

However, I am concerned regarding the entire methodological part of the work. As I wrote, specific scores employed to do the homology modeling are not described.

Even more worrisome is an apparently inadequate process of minimization and equilibration of the models prior to the MD simulations. It seems to be only one cycle done for minimization and equilibration each, with the equilibration of only 2 fs. For such a large molecule this cannot be sufficient. Did the authors investigate the convergence of RMSD in the simulations? What means "the calculated systems have been frozen" (line 153)? Were the systems electro-neutralized? The outline of MD modeling methods suffers from incompleteness and departures from conventional terminology.

Validity of the findings

The results per se might me conclusive and their discussion valid, should there be a confidence that the selection of initial models as well as the results of MD simulations are reliable. Without details of how the initial models have been selected, and without a confidence that proper procedures were used prior doing the production MD runs, the MD simulation results cannot be relied upon.

Additional comments

Although the authors have addressed a potentially very interesting subject, the publication standards of molecular dynamics simulations have not been met. I cannot recommend publication of the manuscript in its present form.

---

## Round 0.2 · accepted · Accept

I have checked your revision. Thank you for addressing critiques of both reviewers.